# Analysis of the Mouse Hepatic Peroxisome Proteome—Identification of Novel Protein Constituents Using a Semi-Quantitative SWATH-MS Approach

**DOI:** 10.3390/cells13020176

**Published:** 2024-01-17

**Authors:** Öznur Singin, Artur Astapenka, Victor Costina, Sandra Kühl, Nina Bonekamp, Oliver Drews, Markus Islinger

**Affiliations:** 1Neuroanatomy, Mannheim Center for Translational Neuroscience, Medical Faculty Mannheim, Heidelberg University, D-68167 Mannheim, Germany; oeznur.singin@medma.uni-heidelberg.de (Ö.S.); tarsistes@gmail.com (A.A.); sandra.kuehl@medma.uni-heidelberg.de (S.K.); nina.bonekamp@medma.uni-heidelberg.de (N.B.); 2Institute of Clinical Chemistry, Medical Faculty Mannheim, Heidelberg University, D-68167 Mannheim, Germany; victor.costina@medma.uni-heidelberg.de (V.C.); oliver.drews@jku.at (O.D.); 3Biomedical Mass Spectrometry, Center for Medical Research, Johannes Kepler University Linz, 4020 Linz, Austria

**Keywords:** peroxisomes, organelle interaction, organelle proteomics, OCIAD1

## Abstract

Ongoing technical and bioinformatics improvements in mass spectrometry (MS) allow for the identifying and quantifying of the enrichment of increasingly less-abundant proteins in individual fractions. Accordingly, this study reassessed the proteome of mouse liver peroxisomes by the parallel isolation of peroxisomes from a mitochondria- and a microsome-enriched prefraction, combining density-gradient centrifugation with a semi-quantitative SWATH-MS proteomics approach to unveil novel peroxisomal or peroxisome-associated proteins. In total, 1071 proteins were identified using MS and assessed in terms of their distribution in either high-density peroxisomal or low-density gradient fractions, containing the bulk of organelle material. Combining the data from both fractionation approaches allowed for the identification of specific protein profiles characteristic of mitochondria, the ER and peroxisomes. Among the proteins significantly enriched in the peroxisomal cluster were several novel peroxisomal candidates. Five of those were validated by colocalization in peroxisomes, using confocal microscopy. The peroxisomal import of HTATIP2 and PAFAH2, which contain a peroxisome-targeting sequence 1 (PTS1), could be confirmed by overexpression in HepG2 cells. The candidates SAR1B and PDCD6, which are known ER-exit-site proteins, did not directly colocalize with peroxisomes, but resided at ER sites, which frequently surrounded peroxisomes. Hence, both proteins might concentrate at presumably co-purified peroxisome-ER membrane contacts. Intriguingly, the fifth candidate, OCIA domain-containing protein 1, was previously described as decreasing mitochondrial network formation. In this work, we confirmed its peroxisomal localization and further observed a reduction in peroxisome numbers in response to OCIAD1 overexpression. Hence, OCIAD1 appears to be a novel protein, which has an impact on both mitochondrial and peroxisomal maintenance.

## 1. Introduction

Peroxisomes are essential, ubiquitous organelles, which are involved in various anabolic and catabolic pathways maintaining cellular lipid homeostasis. Their important role in cellular lipid metabolism is underlined by the existence of numerous inherited peroxisome disorders, which are generally characterized by alterations of the cellular lipid spectrum and which often lead to death during infancy [1]. In addition, peroxisomes perform other less prominent functions in the catabolism of D-amino acids, purines, polyamines and glyoxylate, as well as in antiviral defense [1]. Moreover, to effectively fulfill their functions, peroxisomes do not act as isolated entities inside the cell, but interact with several other subcellular compartments in order to exchange pathway intermediates, transfer membrane lipids or transmit intracellular signals [2]. Hence, peroxisomes can be considered as multifunctional organelles and might perform other undetected functions. Thus, in order to completely understand the significance of peroxisomes for cellular physiology, a thorough annotation of the peroxisomal proteome will help to discover novel functions or further protein constituents of peroxisomal membrane contact sites.

During the last two decades, various proteomics studies increased our knowledge about the peroxisome proteome but also revealed that a significant number of proteins are shared by more than one subcellular compartment [3]. In this regard, it is essential to discriminate mere contaminants from true peroxisome-associated proteins, which can be accomplished by quantitative mass spectrometry (MS), comparing numbers of identified peptides from a protein in individual density-gradient fractions [4]. Liver peroxisomes are spherical-to-tubular organelles, with a diameter between 0.3 and 1.0 µm. In addition to their morphological variation, peroxisomes maintain membrane contacts with different organelles, like the endoplasmic reticulum or mitochondria. As a consequence, peroxisomes sediment across a wide centrifugal force range, and can be found in considerable amounts in mitochondrial- as well as microsome-dominated fractions. Traditionally, peroxisomes have been most successfully isolated using density-gradient centrifugation from the so-called light mitochondrial fraction (LM) [5,6,7]. In order to reduce contamination from microsomes, a pinkish “fluffy layer” sedimenting on top of the LM pellet is usually decanted prior to the final gradient centrifugation [5]. However, a considerable number of peroxisomes can be found in this “fluffy layer” of the LM (FLM). Hence, in this study we aimed at isolating peroxisomes from these two fractions in order to (1) discriminate true peroxisomal constituents from contaminating proteins, (2) unravel whether FLM-peroxisomes (FLM-PO) differ from peroxisomes isolated from the LM (LM-PO) in regard to their proteome; and (3) determine if proteins from attached membrane contact zones might be specifically co-purified with peroxisomes from the two fractions.

Advances in MS technology and bioinformatics during recent years provide increasing sensitivity for protein detection and more reliable approaches for protein quantification. Therefore, more than 10 years after our own publications on the rat peroxisomal liver proteome [7,8], we decided to reassess the proteome of purified rodent liver LM-PO and FLM-PO, using an adapted separation strategy, improved technological equipment and a quantitative SWATH-MS (Sequential Window Acquisition of all Theoretical Mass Spectra) approach for relative quantification, in order to identify still unknown integral peroxisomal proteins but also peroxisome-associated proteins. Compared to conventional quantification strategies, the SWATH approach relies on building predefined spectral libraries and consecutive data-independent acquisition (DIA) of MS spectra by systematically cycling through precursor ion windows [9,10]. This approach leads to acquisition of more complete high-specificity-fragment ion spectra, which enable post-acquisition data analysis of ion-extraction chromatograms, based on multiplexed MS/MS DIA datasets, supporting high quantification accuracy and reproducibility [9].

Subcellular separation in linear density gradients results in a broad, continuous distribution of distinct organelle types peaking at different densities along the gradient slope. Such gradients provide individual organelle fractions of modest purity, and are ideally used for organelle profiling approaches, where protein distribution is quantified across multiple gradient fractions which allow for the allocation of protein constituents to more than a single organelle species [11]. Density gradients, which are shaped to isolate a particular organelle, like, in this case, peroxisomes, are less feasible for gradient profiling experiments but have the advantage of maximum enrichment factors; however, we have to ensure that no other organelles are co-purified with the organelle of interest. Taking this issue into account, we decided to purify peroxisomes from two different prefractions with differing organelle backgrounds. To this end, the so-called light mitochondrial fraction (LM) sedimenting between 2000× g_av_ and 20,000× g_av_ was divided into mitochondria- and microsome-enriched subfractions, which were separately purified on iodixanol gradients of sigmoidal shape. To identify peroxisome-associated proteins, the top and peroxisome-enriched bottom fractions of each of the two gradients were analyzed, using SWATH-MS. Novel proteins, which were consistently enriched in the high-density peroxisome fractions from both gradients, were annotated as potential peroxisomal constituents. For verification of this approach, five proteins—the Platelet-activating factor acetylhydrolase 2 (PAFAH2), the oxidoreductase HTATIP2, the small GTPase SAR1B, the Programmed cell death protein 6 (PDCD6) and the OCIA domain-containing protein 1 (OCIAD1)—were selected for further subcellular localization experiments. From those, PAFAH2, HTATIP2 and OCIAD1 were verified to intrinsically localize at peroxisomes, while PDCD6 and SAR1B localized at sites of the ER, which showed frequent apposition to peroxisomes. Therefore, both proteins might be constituents of ER membrane contact sites co-purified with peroxisomes.

## 2. Materials and Methods

### 2.1. Purification of Peroxisomes

For the purification of peroxisomes, 2-month-old female C57BL/6JRj mice were purchased from Janvier Labs (Le Genest-Saint-Isle, France) and maintained at the animal core facility of the Medical Faculty Mannheim, Heidelberg University, in accordance with the guidelines for care and use of laboratory animals of Germany, at a 12 h/12 h light cycle (animal permit No. I-19/20, Regierungspräsidium Karlsruhe, Germany). All animals were fed ad libitum with standard rodent chow (ssniff Spezialdiäten GmbH, Soest, Germany).

Peroxisome purification was based upon the method described in previous publications [12] and modified as described in the latter (Figure 1). In brief, mice were sacrificed by cervical dislocation and the liver excised after opening of the body cavity. All subsequent steps of the purification procedure were carried out on ice or at 4 °C, using pre-cooled buffers. For each experiment, the livers of 6 mice were pooled to retrieve enough protein in the peroxisome fraction to allow subsequent MS analysis. Subsequently, the livers were rinsed in 0.9% NaCl, immersed in homogenization buffer (HB, 250 mM sucrose, 5 mM MOPS, 1 mM EDTA, 2 mM PMSF, 1 mM DTT, 1 mM ε-aminocaproic acid, pH 7.4) and cut into small pieces. Thereafter, the liver tissue was homogenized using a motor-driven Potter-Elvehjem tissue grinder at 1000 rpm for one stroke in 2 min. After centrifugation at 600× g_av_ for 10 min, the supernatant was aspirated and kept on ice. The remaining pellet was re-homogenized and centrifuged, using the same conditions. The supernatants from both homogenizations were pooled and centrifuged at 2000× g_av_, for 15 min, to produce the “heavy mitochondrial” fraction (HM). After aspirating the supernatant, the mitochondrial pellet was carefully suspended in HB and centrifuged at 2000× g_av_, for 10 min. While the produced pellet was suspended in an appropriate amount of HB for subsequent immunoblot analysis, the supernatants from both centrifuge runs were pooled and centrifuged at 20,000× g_av_, for 20 min. The resulting pellet consisted of an upper, more whitish fluffy layer, which was loosely attached to a brown, more rigid organelle pellet. To separate both layers, the pellet was washed with HB by carefully swirling the centrifuge tube. Subsequently, the floating fluffy layer (FLM) was aspirated with a pipette and thereby removed from the actual “light mitochondrial” pellet (LM). Both the LM and FLM were subsequently suspended in HB and re-centrifuged at 20,000× g_av_ for 15 min. The resulting LM pellet was washed again with HB, as described above, to remove the remaining fluffy layer. Thereafter, both pellets were suspended in 5 mL of HB and subjected to density-gradient centrifugation. Density gradients were prepared in advance by layering iodixanol (Optiprep^®^, Axis Shield, Rodelœkka, Sweden) solutions containing gradient buffer (GB, 5 mM MOPS, 1 mM EDTA, 2 mM PMSF, 1 mM DTT, 1 mM ε-aminocaproic acid, pH 7.4) of 1.26 g/mL (4 mL), 1.22 g/mL (3 mL), 1.19 g/mL (6 mL), 1.15 g/mL (7 mL), and 1.12 g/mL (10 mL) on top of each other. The density of all iodixanol solutions was verified by use of an optical refractometer. Immediately after their preparation, the gradients were snap-frozen in liquid nitrogen and stored at −80 °C until needed. Immediately before the centrifugation, the gradients were thawed in a metal block, which produced the sigmoidal density profile of the gradients (1.12–1.26 g/mL). For peroxisome isolation, the LM and FLM suspensions were layered on top of the gradients, which were centrifuged at 33,000× g_max_ in a VTi50 vertical angle rotor (Beckman, Krefeld, Germany) at an integrated force of 1.256 × 10^6^ g (approx. 38 min). After centrifugation, the gradients exhibited a characteristic, reproducible band pattern (Figure 1), which was used as an orientation for the elution of the gradients. Accordingly, for characterization of the separation via immunoblotting, the gradients were eluted into 5 different fractions (LM1-LM5, FLM1-FLM5), which represented the two peroxisomal fractions LM1 and LM2, an intermediate fraction LM3, the bulk organelle fraction (LM4), and an overlaying zone of floating membranes (LM5). For the proteomics analysis, the highly similar LM1 and LM2 fractions or FLM1 and FLM2 fractions were combined to yield LM-PO or FLM-PO fractions, respectively. Likewise, LM4 and LM5—consisting of about 95% of the total protein (Appendix A) and representing a mixture of the remaining organelles applied to the gradient—were combined, to yield the LM-TOP and FLM-TOP fractions, respectively. After elution, the organelles from the different fractions were pelleted in a final centrifugation step, in order to increase the protein concentration in the fractions and to remove the iodixanol. To this end, the fractions were diluted by 1:4 with HB and centrifuged for 20 min at 30,000× g_av_. The resulting pellets were carefully resuspended in HB, avoiding clumping, and protein concentrations were determined using the Bradford method. While isolation for the immunoblot analysis was repeated thrice, samples from four independent isolation experiments were subjected to the MS analysis.

### 2.2. Immunoblotting

For immunoblotting, 10 µg protein of each organelle fraction was separated in 10% or 12% SDS-polyacrylamide gels (SDS-PAGE), using the discontinuous Laemmli buffer system. After electrophoresis, the gels were blotted onto PVDF membranes, using a semi-dry blotting system (Peqlab, Erlangen, Germany) and a discontinuous buffer system consisting of a 40 mM 6-aminocaproic acid, 25 mM Tris, 20% (*v*/*v*) methanol pH 10.4 cathode buffer, a 30 mM Tris, 20% methanol (*v*/*v*), pH 10.4 anode buffer 1, and a 300 mM Tris, 20% methanol, pH10.4 anode buffer 2 [13]. Western blotting was performed with a current of 0.8 mA/cm^2^ for 90 min. After blotting, membranes were blocked with 5% fat-free milk powder in phosphate buffer saline, Tween (PBST), for 60 min. All antibodies (see Appendix A) were diluted in PBST, 1% FCS. Primary antibodies were incubated overnight at 4 °C, HRP-conjugated secondary antibodies for 1.5 h at RT. In between antibody incubations, the membranes were washed three times with PBST, for 5 min. Antibody signal detection was performed with WesternBright ECL HRP substrate (Advantsa, San Jose, CA, USA) using a Fusion Solo S Western blot imaging system (Vilber-Lourmat, Marne-la-Vallée, France). On some occasions, the blot membranes were probed sequentially by two primary antibodies (see immunoblot originals, in the Appendix A). To this end, after the signal of the first antibody incubation was recorded, the blot was washed overnight in PBST and the next day it was incubated with a second primary antibody, using the same protocol. The molecular weight of the two proteins of interest was carefully selected in a manner which avoided the potential overlap of the two signals.

### 2.3. Sample Preparation for MS Using Gel-Electrophoresis and in-Gel Digestion

SDS PAGE was performed according to the manufacturer’s specifications. In brief, all samples were previously heated for 5 min to 95 °C, cooled on ice, and subsequently loaded onto NuPAGE 4–12% Bis-Tris Gels (Thermo Fisher Scientific, Waltham, MA, USA). To fix the proteins within the polyacrylamide matrix, all the gels were incubated in 5% acetic acid in 1:1 (*v*/*v*) water: methanol, for 30 min. To visualize the protein bands, the slab gels were stained for 60 min with Coomassie R250 (InstantBlue, Expedeon Ltd., Harston, UK), rinsed with water (60 min), and each lane was excised and cut into small pieces.

Subsequently, the proteins were in-gel destained (100 mM ammonium bicarbonate/acetonitrile 1:1 (*v*/*v*)), reduced (10 mM DTT) and alkylated using 50 mM iodoacetamide. Finally, Trypsin digestion was performed using overnight incubation at 37 °C. The peptide-containing supernatants from the Trypsin digestion were collected from the gel pieces. Additionally, the gel pieces were subjected to a peptide extraction step with an acidic (1.5% formic acid) acetonitrile (66%) solution. For each sample, the peptides containing supernatants from both extraction steps were combined and dried down in a vacuum centrifuge.

### 2.4. SWATH-MS Analysis

The dried peptide pellets were re-dissolved in 0.1% trifluoroacetic acid. Subsequently, the peptide solutions were loaded onto a C18 column (Kinetex XB-C18, 150 × 0.3 mm; Phenomenex; Torrance, CA, USA) by direct injection, via an Eksigent Ekspert NanoLC 425 system (AB Sciex, Framingham, MA, USA). Thereafter, peptides were eluted for 125 min with an aqueous–organic 4–48% acetonitrile gradient in 0.1% formic acid, and electrosprayed into a TripleTOF 6600+ mass spectrometer (AB Sciex, Framingham, MA, USA) at a flow rate of 5 µL/min. Each scan cycle consisted of one TOF-MS full scan and up to thirty product ion-dependent (IDA) MS/MS scans of the most intense ions. The mass spectrometer was run in high-sensitivity mode, with dynamic exclusion set to 15 s. All analyses were performed in positive-ion mode.

For generation of the peptide ion library, extracted MS/MS spectra were searched with the ProteinPilot search engine (AB Sciex, Framingham, MA, USA) against a reviewed Uniprot mouse database (April 2021), accepting Cysteine alkylation and common biological modifications. All protein identification experiments were performed with a false discovery rate (FDR) of 1%, using the corresponding decoy database.

SWATH MS was performed according to a customized published protocol [14]. Briefly, using looped 20 Da isolation windows, the SWATH acquisition was performed for an *m*/*z* range of 400–1250 Da. Subsequently, the acquired data were processed with the SWATH Acquisition MicroApp 2.0 in PeakView 2.2 Software (AB Sciex, Framingham, MA, USA), using the spectral ion library generated from the prior data-dependent acquisitions. The following parameters were applied for protein identification in the SWATH mode: 1–4 peptides per protein, 3 transitions per peptide, 99% peptide confidence, 1% FDR, fragment ion extraction window of 5 min, and mass tolerance of 50 ppm. For normalization and analysis of the samples, protein ion intensity data were imported into MarkerView (AB Sciex, Framingham, MA, USA). For normalization, corresponding normalization factors were calculated for each sample, taking into account the total signal intensity of the respective sample after Coomassie staining in the SDS-PAGE gel and its total signal intensity in the MS run. Group differences were examined using the standard *t*-test.

### 2.5. Exploratory Data Analysis

The datasets, comprising protein data for LM-Log2 Fold Change and FLM-Log2 Fold Change, as well as associated UniProt organelle localizations, were loaded into a Python environment and normalized using Min-Max Scaling (Python packages Pandas [15], NumPy [16]). The dataset was subsequently divided into training and testing sets, with 60% of the data allocated for training and 40% for testing (Python package Scikit-Learn [17]). Model Logistic Regression and Random Forest Classifier models were selected to train the model with the dataset. The dataset was further enhanced by the addition of manually curated organelle localizations, based on the information in original publications, to improve the model’s learning capability. Both models were used to predict the ‘Organelle cluster affiliation’ on the test dataset. The complete dataset, along with the top two predictions for each data point, was exported and added to the Appendix A).

### 2.6. Colocalization Experiments Using Overexpression of Myc-Tagged Versions of Candidate Proteins and/or Immunolocalisation of Endogenous Proteins

For the overexpression experiments, pCMV6 entry vectors containing the open reading frames (ORF) of the five candidate mouse proteins mOCIAD1, mPAFAH2, mHTATIP2, mPDCD6 and mSAR1B were purchased from Origene Technologies GmbH (Herford, Germany). To equip the proteins with C-terminal myc-tags, the ORFs were amplified by RT-PCR, using gene-specific primers equipped with appropriate restriction cites (see Appendix A), and recloned into pCMV3A mammalian expression vectors (Agilent Technologies, Santa Clara, CA, USA). The correct insertion and preservation of the ORF was subsequently checked by sequencing (Eurofins Genomics, Ebersberg, Germany).

For the colocalization experiments, HepG2 cells and mouse embryonic fibroblast (for preparation see [18]) were cultured in high glucose Dulbecco’s modified Eagle’s medium (DMEM), supplemented with 10% FBS, 1% penicillin/streptomycin. For immunofluorescence microscopy, the cells were seeded into 24-well cell culture plates (Sarstedt, Nümbrecht, Germany) equipped with Poly-D-lysine-coated (Thermo Fisher Scientific, Waltham, MA, USA) glass coverslips. For overexpression experiments, the cells were transfected on the next day with LipofectamineTM3000 reagent (Thermo Fisher) containing 0.5 μg plasmid DNA per well. After 24 h of incubation, the cells were fixed with 4% paraformaldehyde (PFA) in PBS (pH 7.4) for 20 min, at RT.

### 2.7. Immunofluorescence Analysis

The paraformaldehyde-fixed cells were incubated in a combined blocking/permeabilization solution (1% BSA, 0.2% fish skin gelatin, 0.1% Triton X-100 in PBS. pH 7.4) for 1 h at RT. All primary and secondary antibodies were diluted in blocking buffer (1% BSA, 0.2% fish skin gelatin, 0.1% Triton X-100 in PBS), according to Appendix A. Primary antibodies were incubated overnight at 4 °C, the Alexa dye-labeled secondary antibodies for 1.5 h at RT. In between each antibody incubation step, the cells were rinsed 3 times for 5 min with PBS, at RT. To validate binding specific of the primary antibodies, controls, in which the primary antibody incubation was omitted, were performed for each antibody. Before mounting onto glass microscopy slides, the coverslips were quickly rinsed one more time with ddH_2_O. Afterwards, they were mounted upside down onto the glass slides using Roti, FluorCare immersion medium (Carl Roth GmbH, Karlsruhe, Germany).

Confocal image stacks were acquired, using a C2 Nikon confocal microscope equipped with 488 nm, 561 nm, and 647 nm laser lines and either an ApoPlan 60× (oil immersion, 1.4 NA) or an ApoPlan 100× (oil immersion, 1.45 NA) objective (Nikon Europe B.V., Amstelveen, The Netherlands). The thickness of single optical sections was set to 0.5 µm in stacks of 10–20 µm total depth. Image resolution was set to 1024 × 1024 pixels, with a fixed 0.08 μm pixel size. The open-source software Fijii ImageJ 1.54f was used for all post-imaging analysis. For the quantification of cell numbers, stacks of images were merged into a maximum-intensity projection. For the quantification of peroxisome densities, single planes from the center of the cell were used. Counting, size and relative area calculation, and circularity determination of fluorescent signals in HepG2 WT were performed with Fiji ImageJ, using automated thresholding and the “analyze particles” command (size: 2 pixels-infinity, circularity: 0.0–1.0). The mitochondrial network analysis was performed with the ImageJ MiNA plug-in. For statistical analysis of the organelle morphology changes in the protein overexpression experiments, a one-tailed, unpaired *t*-test was applied (* *p* < 0.05; ** *p* < 0.01; *** *p* < 0.005; **** *p* < 0.001; ns: not significant) using the GraphPad Prism 10.0.2 software (Dotmatics Software Inc., Boston, MA, USA). Sample sizes used for data analysis are indicated under the corresponding figures. All quantitative data were collected from at least three independent experiments.

## 3. Results

### 3.1. Mouse Liver Peroxisomes Can Be Isolated in Comparable Quality from Different Prefractions

Prior to subjecting density gradient fractions to the MS analysis, we first aimed to characterize the LM and FLM, as well as corresponding fractions from the density gradients, by immunoblotting. Since the FLM is mechanically washed off the LM, we first evaluated the extent to which the organelle composition of the LM and FLM differs, and if both the LM and FLM fraction can be reproducibly separated from each other. To this end, the fractions obtained from differential centrifugation from four independent isolation experiments were analyzed using immunoblotting (Figure 2A). The peroxisomal proteins catalase, PXMP2, ABCD3, ACOX1, ACAD11, SCP-X and PEX3 were consistently and significantly enriched in both the LM and FLM fraction, when compared to the heavy mitochondrial fraction (2700× g_max_ pellet, HM) and the microsomal fraction (100,000× g_av_ pellet, Mic) (Figure 2A). Of note, according to the protein band intensities, the peroxisomal proteins show a comparable abundance in both LM and FLM, and no protein-specific differences were observed between peroxisomes contained in the LM and FLM. As expected, mitochondria represented by ATPA, ATPB5 and VDAC1 exhibit their highest concentration in HM and decrease continuously from LM to FLM to MIC. In contrast, the ER proteins GRP78/BiP and FATP4 reproducibly increase in concentration from HM to MIC. In summary, the results from the immunoblots reveal that both the LM and FLM are comparably enriched in peroxisomes but are differently contaminated with other organelles; the LM to a higher extent with mitochondria, and the FLM more with microsomal vesicles. Hence, we decided to isolate peroxisomes from both prefractions in order to analyze whether (1) a comparable set of unknown proteins are enriched in both the LM and FLM peroxisomal fractions, suggesting those as candidate peroxisome proteins, and (2) whether a distinct subset of proteins from either mitochondria or the ER are enriched in either the LM or FLM peroxisomal gradient fractions, which might indicate that specialized domains from both organelles are co-purified.

First, in order to compare whether peroxisomes can be isolated to comparable purities from both prefractions, sigmoid-shaped 1.12–1.26 g/mL iodixanol gradients were loaded with either the LM or FLM from the same experiment, centrifuged in parallel, and analyzed using immunoblotting. Visually, the gradients showed a similar band pattern after the centrifugation, presenting with two distinct bands at the bottom and the bulk of organelle material (approx. 95% of the total protein) at the top of each gradient. To evaluate the separation of individual organelles, the LM and FLM gradient were equally eluted into five fractions, as depicted in Figure 1. Immunoblots show that peroxisomes according to the peroxisomal marker proteins ACBD5, catalase, PEX3, PXMP2, ABCD3 and ACAD11 are most strongly enriched in the high-density fractions LM1/FLM1 and LM2/FLM2, which correspond to the two distinct bands at the bottom of the LM and FLM gradients, respectively (Figure 2B). Notably, band intensities of the peroxisomal marker proteins do not significantly differ between LM1 and FLM1 or LM2 and FLM2 fractions, indicating that peroxisomes from both fractions are of comparable purity. Moreover, a comparison with the band intensities of the peroxisomal proteins in the PNS reveals the strong enrichment of peroxisomes in the LM1/FLM1 and LM2/FLM2 fractions. Indeed, according to enzymatic activity measurements, the protocol used in this work reproducibly allows for the isolation of peroxisomes with a purity above 95% for the LM1 and 90% for the LM2 fraction [7,19].

In contrast to differential centrifugation, mitochondria exhibit a significantly lower density than peroxisomes in iodixanol gradients. Consequently, the mitochondrial proteins ATPA and VDAC1 enrich at lower densities between LM3 and LM5 (Figure 2B). Of note is the fact that signals for both proteins are considerably less intense in the corresponding FLM3–FLM5 fractions, which is in line with the lower percentage of mitochondria in the FLM. The ER resident proteins GRP78 and ERP29 increasingly accumulate from LM1/FLM1 to LM5/FLM5 fractions, respectively, which demonstrates that microsomes are only a minor contaminant in the peroxisome-enriched fraction from both the LM and FLM prefraction. Interestingly, the acyl-CoA synthase ACSL1, which was localized to the ER, mitochondria and peroxisomes [20], shows the most prominent signal in the microsomal prefraction, indicating that ACSL1 is predominantly localized at the ER. However, after density-gradient centrifugation, ACLS1 signals are most intense in the peroxisome-enriched LM1/2 and FLM1/2 fractions, implying that a minor fraction of the protein is likely localized at the peroxisomes. Likewise, the tethering protein VAPB, which facilitates membrane contacts to peroxisomes [21,22], does not follow the typical microsomal enrichment pattern. While its strongest signals, in line with its ER localization, occur in the microsomal prefraction, VAPB accumulates at higher densities, comparable to peroxisomal proteins. Hence, proteins from membrane contacts appear to be co-purified with peroxisomes, which might allow their identification as peroxisome-associated proteins. The results from the immunoblots demonstrated that peroxisomes can be isolated with similar quality from the classic LM prefraction, as well as the overlaying FLM fraction. Accordingly, true peroxisomal proteins, as well as proteins, which are in physical contact with peroxisomes such as the constituents of organelle contact sites, should be consistently enriched in peroxisomes fractions purified from the LM, as well as the FLM prefraction. Nevertheless, immunoblotting results are always limited by the quality of the primary antibodies available and, hence, a more extensive characterization of the individual fractions on a large scale using MS proteome analysis can be used to confirm the results gained from immunoblotting.

### 3.2. SWATH MS Analysis of Purified Peroxisomes from LM and FLM Prefractions Result in a Highly Similar Enrichment of a Subset of Proteins

With respect to the results from the immunoblot analysis, we considered to exploring the resolving power of the sigmoid-shaped density gradients, in order to define the mouse liver peroxisome proteome. As shown by immunoblotting, proteins with an association with peroxisomes can be, irrespective of the organelle composition of the prefraction, separated from the mass of organelles, which are retained at the top of the gradients (Figure 2). Accordingly, both the LM and the FLM fractions from four independent isolation experiments were separated on respective sigmoid-shaped 1.19–1.26 g/mL density gradients. Subsequently, from each experiment, the TOP-factions (LM-TOP/FLM-TOP), containing the bulk organelle material applied to each gradient and the high-density peroxisome fractions (LM-PO/FLM-PO) was analyzed using quantitative SWATH-MS. After the generation of a spectral peptide ion library through data-dependent acquisition, DIA SWATH-MS acquisition was performed for each sample, and peak areas of peptide fragment ions were quantified. Data were normalized according to total signal intensity of each sample separated by SDS-PAGE and subsequent Coomassie staining, as well as total MS signal intensity.

Overall, 1071 proteins were quantified in all four analyzed sample groups and subjected to additional downstream analysis (Appendix A). Generally, protein quantifications are based on at least three replicate values for ≥90% of the analyzed proteins (Appendix A), underlining the high data completeness in SWATH-MS compared to classic shotgun proteomics approaches. To further evaluate the data, function and subcellular localization of the individual proteins were annotated, according to the information available at UniProt. Additionally, all proteins with a proposed peroxisomal localization or a localization to peroxisomes and one further compartment were hand curated, according to the information from the original publications. First, in order to compare whether peroxisomes can be isolated to comparable purities from both prefractions, sigmoid-shaped 1.12–1.26 g/mL iodixanol gradients were loaded with either the LM or FLM from the same experiment, centrifuged in parallel, and analyzed using immunoblotting. A comparison of the 1071 proteins in the TOP-fractions from the LM- and FLM gradients illustrates the differing organelle composition of the LM and FLM fraction (Figure 3, Appendix A) and confirms the result from the immunoblot analysis. While nearly all identified microsomal and ribosomal proteins are present in higher numbers in FLM-TOP, mitochondrial and lysosomal proteins are more abundant in LM-TOP (Figure 3A–C).

Importantly, a comparison of the corresponding LM-PO and FLM-PO does not show a similar distribution of the mitochondrial and microsomal proteins (Figure 3D–F). Indeed, proteins from both organelles inconsistently differ in abundance between both peroxisome fractions, indicating that these proteins do not distribute in LM-PO and FLM-PO according to the organelle composition of the LM and FLM prefractions, as would be expected for mere contaminants. Of note, proteins with a published peroxisomal localization do not differ significantly in abundance between both fractions (average ratio in LM-PO/FLM-PO = 1.1). In order to define the group of proteins which were enriched in the peroxisomal fractions, protein quantifications between LM-TOP and LM-PO, as well as FLM-TOP and FLM-PO, were compared. Consistently, most proteins with a published peroxisomal localization were found to be enriched in the respective peroxisomal fractions (average enrichment factor for LM-PO = 10.37, FLM-PO = 7.16) (Figure 4A,B). Only 1% and 2% of the peroxisomal proteins described to date were not significantly enriched in LM-PO and FLM-PO fractions, respectively. A closer look at the identity of these proteins (UGT1A1, ACTG1, SOD2, SLC27A2, MGST1, CYB5A, FABP1, and CYB5R3 for LM; UGT1A1, RHOA, CYB5A, ACTG1, SLC27A2, MGST1, HSD3B3, CYB5R3, FABP1, RAB10, SOD2, RAB14, and ALDH3A2 for FLM) reveals that these are uniformly proteins which still lack profound experimental confirmation of their peroxisomal localization [3]. Moreover, all are, with respect to the UniProt database, localized at a second subcellular compartment and, hence, might be found in similar concentrations in PO and TOP fractions. As expected, the bulk of mitochondrial, ER, ribosomal and lysosomal proteins were observed to be enriched in LM-TOP and FLM-TOP. However, a much smaller proportion of proteins previously localized to mitochondria, the ER and lysosomes, were not in enriched in the gradient TOP, but in the PO fraction from both the LM and FLM gradients. Notably, the proportion of these significantly enriched mitochondrial, lysosomal and ER proteins was found to be comparable between LM-PO and FLM-PO (Figure 4C,D), thus suggesting that this protein subset appears to include a large percentage of candidates localizing to peroxisomes, in addition to other organelles.

To compare whether both the LM and FLM density gradients lead to a similar enrichment profile of individual proteins, a regression of FLM-TOP/FLM-PO vs. LM-TOP/LM-PO protein abundance ratios is shown in Figure 5 (the positioning of individual proteins in the plot is accessible in Appendix A). As already shown in the volcano plots (Figure 3A,D), the bulk of the mitochondrial and ER/ribosomal proteins are enriched to a different extent in LM-TOP and FLM-TOP fractions, thus creating two prominent protein clusters in the scatter plot of Figure 5.

The annotated peroxisomal proteins form, according to their enrichment in the LM-PO and FLM-PO fractions, a third cluster at the opposite quadrant of the graph. Importantly, a linear regression for the peroxisomal proteins demonstrates their similar enrichment in LM-PO and FLM-PO (R^2^ = 0.8; Figure 5). Based on the log2 LM and FLM ratios, a machine learning-based prediction model was trained in order to associate the individual proteins with the clusters “peroxisomes”, “mitochondria”, “ER”, and “multilocalized” proteins, accessible in Appendix A. Notably, several proteins like ABHEB, ACNT1, LACB2, and PMVK, which have been suggested for a peroxisomal localization [3], can be found in the cluster of the bona fide peroxisomal proteins, supporting their localization to the organelle (Figure 5). By contrast, several other proteins, which have been previously reported for a peroxisomal localization, locate plainly among the ER (NUD11, ACSL5, CYB5A) or mitochondrial protein cluster (THIL, SOD2, SLC25A17), implying that they are more likely contaminating proteins in peroxisome-enriched fractions. In addition to these three organelle clusters, a significant number of proteins showed no significant enrichment in either the LM/FLM-TOP or LM/FLM-PO fraction, thus grouping around the crossing of the *y*- and *x*-axis. A closer look at the nature of these proteins reveals that these are most likely not arbitrarily located in between the peroxisomal and mitochondrial/ER clusters: among the proteins are several proteins which were already described as being dually localized at both peroxisomes and mitochondria (e.g., MAVS, MIRO1) or peroxisomes and the ER (e.g., ACSL1, AL3A2). Moreover, several small GTPases, which seem to play a role in regulating peroxisomal processes [23], locate in this region (RAB14, RAB18, RAB10), but there are also the ER-tethering proteins VAPA and VAPB, which interact with peroxisomal tail-anchored proteins ACBD5 [21,22]. According to their location among the described organelle clusters, further hitherto uncharacterized proteins may be explored as candidates for a peroxisomal localization or to confirm their exclusive localization to mitochondria or the ER.

### 3.3. Overexpression of Selected Candidates Suggests That the Fractions of Purified Peroxisomes Contain Intrinsic Peroxisomal Proteins and Proteins Associated via Co-Purified Membrane Contacts

To validate this concept, five candidate proteins, which all significantly accumulated in the peroxisomal cluster, were selected for further localization experiments. These included the oxidoreductase HIV-1 Tat interactive proteins 2 (HTATIP2), the serine esterase Platelet-activating factor acetyl-hydrolase 2 (PAFAH2), which both contain a predicted PTS1 sequence, the Secretion-associated RAS superfamily GTP-binding protein 1B (SAR1B), the Programmed cell death protein 6 (PDCD6) and the Ovarian carcinoma immunoreactive antigen domain-containing protein 1 (OCIAD1). To analyze their intracellular localization, the corresponding mouse cDNAs of the selected candidates were cloned into expression vectors containing N-terminal myc-tags, which were transfected into HepG2 cells and examined using confocal immunofluorescence microscopy. As shown in Figure 6A,B, the signals for the PTS1-containing proteins HTATIP2 and PAFAH2 co-localized almost entirely with the PEX14 antibody signal, thus confirming their specific targeting of peroxisomes.

By contrast, a direct localization of SAR1B, as well as PDCD6, to peroxisomes could not be observed (Figure 6C,D). As expected by its known function in vesicle formation at ER exit sites (ERESs) [24], SAR1B signals highlight a typical reticular network, characteristic of the ER. However, the overlay with the peroxisomal PEX14 signals reveals an intense association between peroxisomes and the frequent focal accumulations of SAR1B signals among the ER. Transepts through the image stack in the Z-plane show that SAR1B and PEX14 signals emerge from the same intracellular planes. A comparable relationship between peroxisomes and SAR1B-positve ER sites was also observed by antibody-staining of the endogenous SAR1B (Appendix A). Thus, in HepG2 cells, ERESs appear to stay at ER sites, which are in close proximity to the ER. Interestingly, in hepatocytes, the so-called wrapper membrane contacts between the rER and peroxisomes have been described recently [25]. In these membrane contacts, peroxisomes, and also mitochondria, are virtually surrounded by sheets of the rER, which might explain why SAR1B as a component of the ER secretory machinery was co-purified with peroxisomes. PDCD6 is a calcium-binding protein, which acts a bridge for proteins in several protein complexes. Therefore, it has been reported to localize at several intracellular compartments, including endosomes and ERESs [26,27,28,29]. Expression of myc-PDCD6 in HepG2 cells resulted in a spot-like intracellular staining pattern, which could not be identified as peroxisomes (Figure 6D). However, as observed for SAR1B, PEX14 and PDCD6, signals frequently exhibited partial colocalization, which might indicate an interaction of peroxisomes with PDCD6 localized at another intracellular compartment. Since the vesicle-like staining for PDCD6 might point to a localization at endosomes, RAB5A and LAMP1 antibodies were used to mark the early- and late-endosomal compartment. Signals from both antibodies, however, also did not show a high degree of overlap. SAR1B, as well as VAPB, has been described as accumulate at ERESs. Therefore, myc-PDCD6 signals were compared with the staining pattern of both endogenous proteins (Appendix A). Both SAR1B and VAPB were visualized in patterns characteristic of the rER. As expected, the vesicle-like signals for PDCD6 did not completely colocalize with either VAPB or SAR1B. However, the spotted PDCD6 signals characteristically accumulate in areas with high SAR1B or VAPB intensities. Hence, they might indicate early-export vesicles budding from ERESs. Taken together, both PDCD6 and SAR1B might be ERES components, which are associated with peroxisomes via wrappER membrane contacts.

### 3.4. OCIAD1 Localizes to Peroxisomes and Mitochondria and Modulates the Morphology of Both Subcellular Compartments

OCIAD1/Asrij is a membrane protein, which localizes at the inner membrane of mitochondria [30]. However, its additional peroxisomal localization was recently proposed after an OCIAD1 bait identified several peroxisomal membrane proteins in a BioID screen [31]. Overexpression of OCIAD1 in HepG2 cells confirmed the dominant mitochondrial localization and also its targeting of peroxisomes (Figure 7A,B). Likewise, in mouse embryonic fibroblasts, endogenous OCIAD1 exhibited an intense staining of mitochondria, while smaller vesicle-like OCIAD1 signals colocalize to peroxisomes marked by PEX14 (Figure 7D). At mitochondria, increasing OCIAD1 levels have been described to promote mitochondrial fractionation, in order to reduce mitochondrial OXPHOS activity [32]. Confirming these observations, we observed a reduction in mitochondrial branch length in response to OCIAD1 expression (Figure 7C). Remarkably, OCIAD1 expression showed an impact on peroxisomes, as well. Increased OCIAD1 abundance significantly reduced cellular peroxisome numbers and particle size (Figure 7C). In addition, an increase in the average peroxisomal circularity index implies a reduction in tubular peroxisomes, which are regarded as elongating precursors in the process of peroxisome division [33]. Notably, in the HepG2 cells with the highest OCIAD1 signal intensities, PEX14 was observed to also localize to the mitochondrial network, but not, or less, to spherical peroxisomal structures (Appendix A). Indeed, PEX14 has been observed to be targeted to mitochondria in mammalian cells, when peroxisomes are absent [34], suggesting that OCIAD1 has a significant impact on peroxisomal maintenance. Moreover, upon OCIAD1 expression, spherical peroxisomes align to a higher extent with mitochondria (Appendix A), which might indicate that both organelles physically interact in order to transmit signals regulating each other’s metabolism or physiological state. In this regard, we conclude that OCIAD1 is another protein colocalizing at mitochondria and peroxisomes to coordinate and link the metabolic activities of both organelles, meriting being explored by future investigations.

## 4. Discussion

During the last two decades, several proteomics studies have been published, which were undertaken to characterize the proteome of liver and kidney peroxisomes [7,8,23,35,36,37,38]. During recent years it became also more and more evident that a high proportion of proteins are localized to more than one subcellular compartment [39], and that organelles dynamically interact with each other via physical membrane contacts [40], changing our view on co-purified proteins, which previously were mostly regarded as mere contaminants. Technical and bioinformatics innovations continuously improve the detection limits and precision of quantitative mass-spectrometry-based proteomics. In this work, we therefore reassessed the rodent liver peroxisomal proteome using a SWATH-MS approach in order to identify additional, low-abundant peroxisomal or peroxisome-associated proteins from potential co-purified contact sites. Protein correlation profiling is a powerful technique, and is the gold standard for annotating a complete cellular proteome with respect to the different subcellular locations [41,42]. However, as it requires comparative MS analysis of numerous subcellular fractions, it is intensive in terms of time, work and computation. Thus, we explored an alternative approach for an organelle-centered survey in order to define the peroxisomal protein composition: instead of a continuous linear density gradient, a sigmoidal gradient with a steep incline in density in the middle of the gradient was used. Such a gradient allows for the efficient separation of peroxisomes, as the organelles of interest from the remaining subcellular components. Accordingly, only two fractions—the peroxisome-enriched and the bulk organelle Top-fraction were subjected to quantitative SWATH-MS analysis. Applying this gradient fractionation scheme to two different prefractions with distinct organelle compositions allowed us to construct a 2-dimensional coordinate system, based on the log2-ratios between LM-TOP/LM-PO and FLM-TOP/FLM-PO. Such a diagram allows for the association of proteins into four different main clusters: peroxisomes, mitochondria, microsomes, and a cluster of multilocalized proteins found at peroxisomes and a second subcellular compartment. Performed as an initial study, we were able to quantify 1071 proteins identified in the four different fractions. A more extensive protein and peptide fractionation, prior to the MS analysis, will further increase the number of proteins, which can be categorized according to their peptide peaks areas, and will accordingly allow for identifying additional low-abundance peroxisome-associated proteins.

As a result of this proteomics study, we were able to associate a considerable number of proteins as candidates for an intrinsic peroxisomal localization, or as candidate peroxisome-associated proteins, which may originate from co-purified membrane contact sites or temporary interaction with peroxisomal membrane protein complexes. From this set of potential candidates, the five proteins HTATIP2, PAFAH2, SAR1B, PDCD6 and OCIAD1 were selected for further validation. Hence, it is tempting to speculate on the potential peroxisome-related roles of these proteins. With the two PTS1-containing enzymes, HTATIP2 and PAFAH2, we confirmed the peroxisomal localization of two additional matrix proteins, which may complement the metabolic repertoire of peroxisomes. HTATIP2 was originally described as a tumor suppressor, and HIV1-binding protein with intrinsic serine/threonine kinase activity [43]. HTATIP2 crystallization, however, revealed that the enzyme belongs to the short-chain dehydrogenase superfamily, possessing a hydrophobic fold for potential lipid binding [44]. While the physiological substrate of the enzyme has not yet been identified, arachidonic acid-binding was described for an HTATIP2/ACSL4-containing protein complex [45]. Moreover, overexpression of HTATIP2 in HepG2 cells was recently reported to decrease β-oxidation of palmitate and increase the corresponding lipid droplet formation, thus linking the enzyme’s function to lipid metabolism [46]. However, future studies are required to unravel the role of HTATIP2 in peroxisome metabolism. PAFAH2 is a hydrolase, which was described as being able to cleave a variety of acyl-chains from phospholipids, thus functioning as a phospholipase [47]. More recently, it was shown that PAFAH2 preferentially removes oxidized phospholipids, fulfilling a protective role under oxidative stress [48]. Since the numerous peroxisomal oxidases continuously produce H_2_O_2_, phospholipid oxidation of the inner leaflet of the peroxisome membrane may be a major threat to maintaining physiological membrane properties of the organelle. In this regard, HTATIP2 might remove oxidized fatty acids from phospholipids of the peroxisome membrane, in order to guarantee proper membrane stability and dynamics.

While the two PTS1-containing enzymes are unambiguously targeted at peroxisomes, the targeting analysis showed that the two ER-associated proteins SAR1B and PDCD6 are most likely not intrinsic proteins of peroxisomes. However, peroxisomes are in frequent membrane contact with ER, including the smooth ER and, in hepatocytes, also the rough ER [18,25]. While contacts with the sER may preferentially facilitate the transfer of phospholipids from the ER to peroxisomes and the exchange of lipid metabolites between both organelles [49], a functional relationship behind the contacts between the rER and peroxisomes is less obvious. A recent publication reported that so-called wrappER membrane contacts between the rER and mitochondria, as well as peroxisomes, are involved in the regulation of lipid loading onto export lipoproteins, since both organelles can reduce the lipid flow towards the exported lipoproteins by the breakdown of fatty acids via β-oxidation [25]. In mammals, SAR1B is, in contrast to SAR1A, the GTPase which specifically initiates the COPII-coat assembly of vesicles for the export of blood-circulating lipoproteins at the ERES [50]. Likewise, PDCD6, also known as ALG-2, is a calcium-binding protein, which is recruited to ERESs via binding to SEC31, in order to stabilize COPII coat assembly [51,52]. Thereby, PDCD6 may assist in producing large vesicles at the ERES, which are require to export large cargo such as procollagen or lipoproteins [53]. Thus, the co-purification of both proteins with peroxisomes might result from a tissue-specific cooperation between peroxisomes and the rER in liver, via membrane contacts, which is required for the routing of lipid flow toward extrahepatic tissues. Recently, with ARF1, another small GTPase with a known role in vesicle coat assembly has been reported to reside at membrane contact between peroxisomes and mitochondria, mitochondria and lipid droplets, as well as peroxisomes and lipid droplets, thereby regulating cellular lipid flux [54]. Moreover, since PDCD6 is an adapter protein, which stabilizes protein complexes not only at the ERES but also at other cellular locations [55], it cannot be excluded that both SAR1B and PDCD6 interact transiently with peroxisomal membrane proteins to directly influence peroxisomal metabolism or maintenance.

The transmembrane protein OCIAD1 was initially identified as a protein which is highly expressed in embryonic stem cells and cardiovascular lineages, while it is down-regulated in many differentiated cell types [56]. Much later, OCIAD1 was identified as a constituent of the inner mitochondrial membrane, where it interacts with Complex I and Complex III of the electron transport chain and down-regulates mitochondrial oxidative phosphorylation (OXPHOS) in embryonic stem cells [30,57]. Thereby, embryonic stem cells appear to be maintained in a glycolytic state, while suppression of OCIAD1 expression supports the metabolic switch to OXPHOS typical of differentiated cells [57]. Several reports showed that fused mitochondrial networks are found in cells depending mainly on OXPHOS, while cells with spherical mitochondria have a tendency to rely on glycolysis for energy production [58]. In line with this, increased OCIAD1 expression was found to promote mitochondrial fission by acting in concert with the obligate regulators of mitochondrial dynamics DRP1 and MFN, thereby resulting in a less-elaborate mitochondrial network, whereas OCIAD1 depletion elongates mitochondria [57,59]. The peroxisomal localization of OCIAD1 was recently proposed after the interaction of OCIAD1 with several peroxisomal proteins was revealed in a BIOID screen, performed to define the mitochondrial interactome [31]. Here, we confirm the peroxisomal localization of OCIAD1 in mice by (1) enrichment in purified peroxisome fractions, (2) targeting of overexpressed mOCIAD1 to peroxisomes and (3) immunofluorescence localization of endogenous OCIAD1 in mouse embryonic fibroblasts. Moreover, for the first time, we give evidence that OCIAD1 levels not only influence mitochondrial dynamics but also affect peroxisome maintenance. Increased OCIAD1 levels induce fragmentation of the mitochondrial network, likely in order to inhibit OXPHOS, and, in parallel, reduce peroxisome abundance. Physiologically, such a coordinated response would ensure that acetyl-CoA and octanoyl-CoA produced from peroxisomal β-oxidation would accumulate in the cell under conditions where acetyl-CoA cannot enter mitochondrial citrate, since FADH_2_ and NADH are not transferred to the mitochondrial OXPHOS chain. Since several of the key factors of mitochondrial fission, such as DRP1, MFF and FIS1 are shared between peroxisome and mitochondria [33], it will be tempting to analyze how OCIAD1 might be integrated into the protein network coordinating the abundance of both organelles, potentially by blocking peroxisomal fission or as an alternative, inducing peroxisomal degradation by autophagy.

In summary, this work introduces a complementary and efficient approach for using quantitative proteomics in order to define the proteome of peroxisomes more accurately. Comparing highly purified peroxisome fractions with the bulk organelle fraction of a sigmoid-shaped density gradient allowed for the grouping of the quantified proteins into distinct organelle panels, thereby revealing several novel candidates for a peroxisomal localization or association. Here, we subsequently validated the localization of five selected proteins from the candidate list and described their association with the peroxisomal compartment. Successive experiments have to be performed to validate the localization of remaining peroxisomal candidate proteins from this study, and to further unravel the role of HTATIP2, PAFAH2, SAR1B, PDCD6 and OCIAD1 at peroxisomes.

## Figures and Tables

**Figure 1 cells-13-00176-f001:**
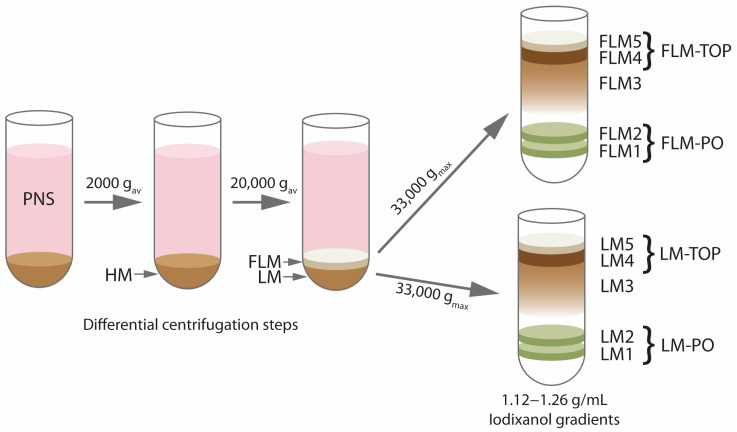
Scheme illustrating the major fractions generated during peroxisome purification, as performed in this work. For a detailed description of the separation protocol, see the Section 2 of the manuscript. Abbr.: PNS—post nuclear supernatant, HM—heavy mitochondrial fraction, LM—light mitochondrial fraction, FLM—“fluffy” layer of the light mitochondrial fraction, LM/FLM1–5: fractions of the corresponding density gradients analyzed using immunoblotting, LM/FLM-TOP, LM/FLM-PO—respective fractions analyzed using SWATH-MS.

**Figure 2 cells-13-00176-f002:**
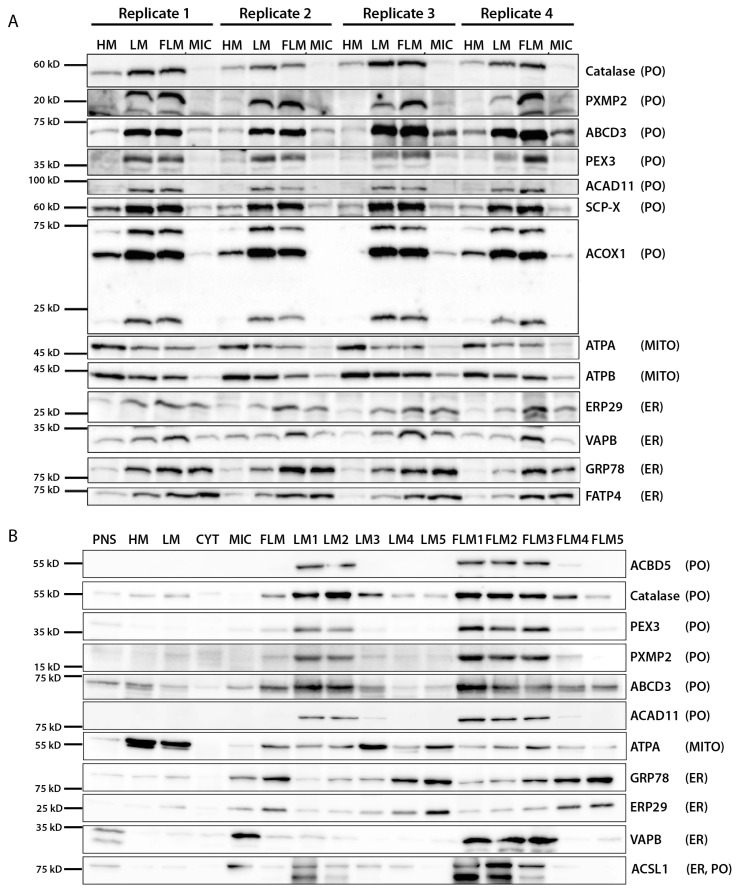
Immunoblot analysis of the main fractions generated by differential or gradient centrifugation during peroxisome purification, as performed in this work. (**A**) Distribution of selected organelle marker proteins in the main fraction generated by differential centrifugation (four independent experimental replicates). (**B**) Distribution of selected organelle marker proteins in all fractions generated during the purification protocol in an exemplary experiment. Abbreviations: PNS—post nuclear supernatant, HM—heavy mitochondrial fraction, LM—light mitochondrial fraction, FLM—“fluffy layer” from LM, CYT—cytosolic fraction, MIC—microsomal fraction, PXMP2—Peroxisomal membrane protein 2, ABCD3—ATP-binding cassette sub-family D member 3, ACAD11—Acyl-CoA dehydrogenase family member 11, SCP-X—Sterol carrier protein X, ACOX1—Acyl-CoA oxidase 1, ATPA—ATP synthase subunit α, ATPB—ATP synthase subunit β, ERP29—Endoplasmic reticulum resident protein 29, VAPB—Vesicle-associated membrane protein-associated protein B, GRP78—Endoplasmic reticulum chaperone BiP, FATP4—Long-chain fatty acid transport protein 4, ACSL1—Long-chain-fatty-acid-CoA ligase 1.

**Figure 3 cells-13-00176-f003:**
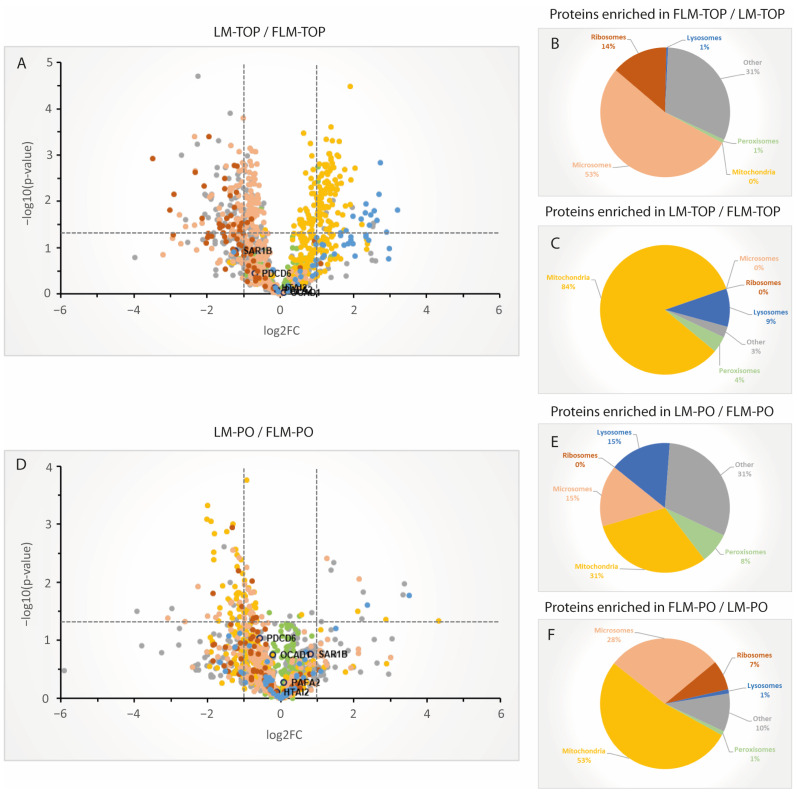
Organelle distribution of proteins quantified in the fractions analyzed using SWATH-MS. (**A**) Log2-scaled volcano plot depicting the quantification ratios for the individual proteins between LM-TOP and FLM-TOP. Proteins were color-coded, as indicated in (**B**). (**B**) Organelle annotation of the proteins shown in (**A**) (in %) significantly enriched in the FLM-TOP compared to the LM-TOP fraction (*p* < 0.05). (**C**) Organelle annotation of the proteins (in %) significantly enriched in the LM-TOP compared to the FLM-TOP fraction (*p* < 0.05). (**D**) Log2-scaled volcano plot depicting the quantification ratios of the individual proteins between LM-PO and FLM-PO. (**E**) Organelle annotation of the proteins (in %) significantly enriched in the LM-PO compared to the FLM-PO fraction (*p* < 0.05). (**F**) Organelle annotation of the proteins (in %) significantly enriched in the LM-PO compared to the FLM-PO fraction (*p* < 0.05).

**Figure 4 cells-13-00176-f004:**
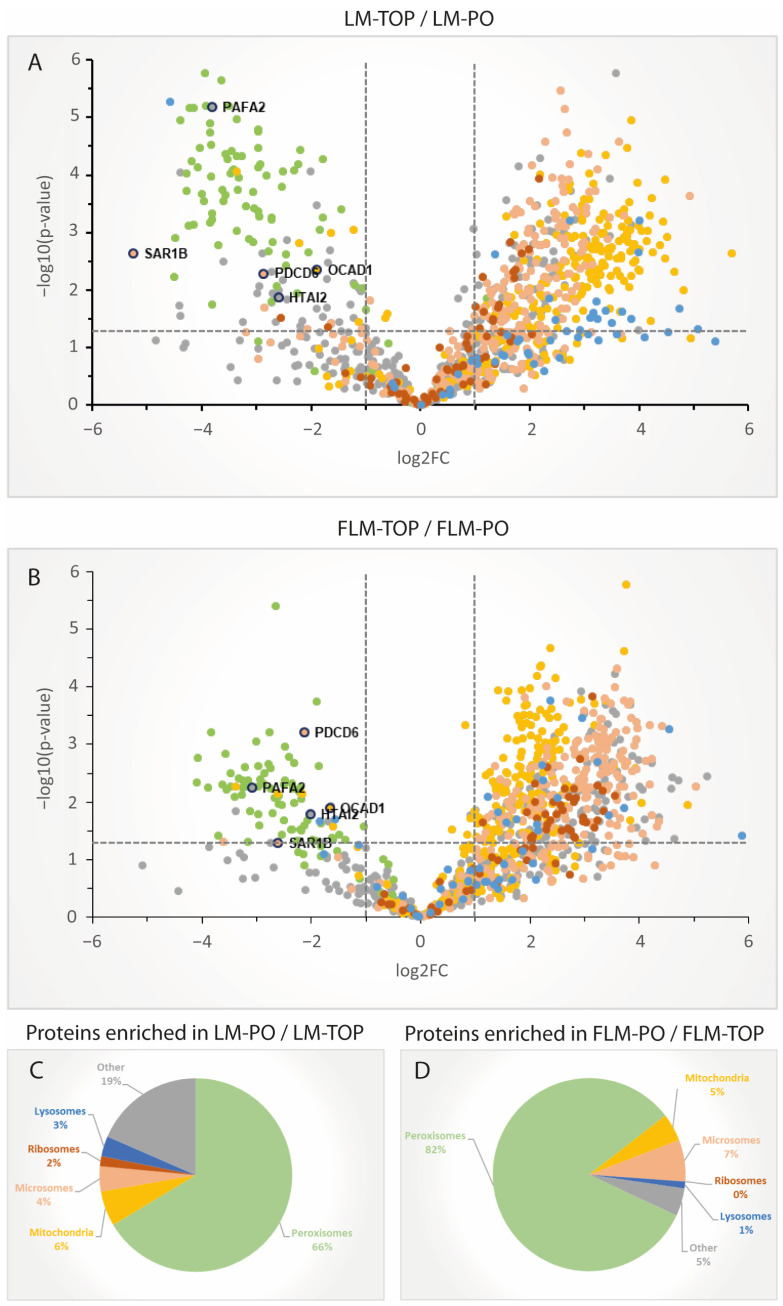
Organelle distribution of proteins quantified in the fractions analyzed using SWATH-MS. (**A**) Log2-scaled volcano plot depicting the quantification ratios for the individual proteins between LM-TOP and LM-PO. (**B**) Log2-scaled volcano plot depicting the quantification ratios of the individual proteins between FLM-TOP and FLM-PO. (**C**) Organelle annotation of the proteins (in %) significantly enriched in the LM-PO compared to the LM-TOP fraction and (**D**) FLM-PO compared to the FLM-TOP fraction (*p* < 0.05), respectively.

**Figure 5 cells-13-00176-f005:**
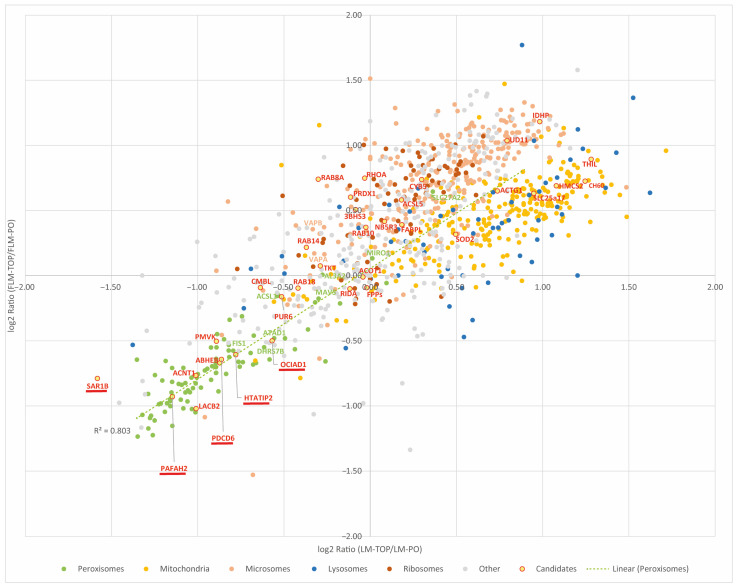
Scatter plot depicting the ratios of the proteins between LM-TOP/LM-PO (*x*-axis) and FLM-TOP/FLM-PO (*y*-axis) on a log2 scale. Note that most proteins assemble according to their organelle localization, in three characteristic clusters. In contrast, proteins which were localized in both peroxisomes and mitochondria/ER tend to assemble near the center of the graph, according to their similar abundance in PO and TOP fractions. Proteins which were recently suggested for a peroxisomal localization but which require further confirmation according to Yifrach et al., 2018 [3] are highlighted (red); peroxisome-associated proteins which were identified and validated in this work are additionally underlined.

**Figure 6 cells-13-00176-f006:**
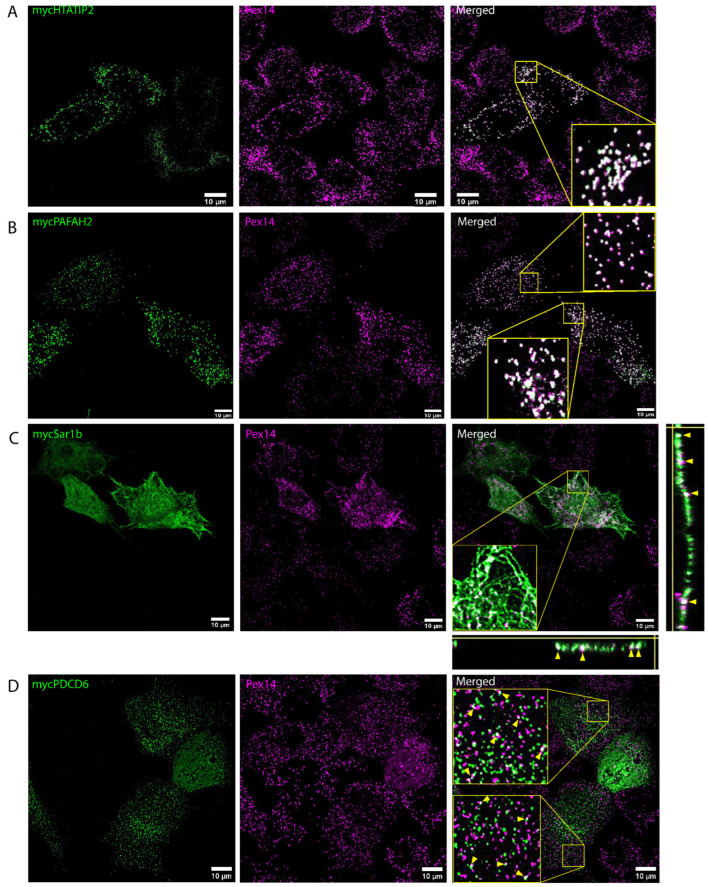
Confocal immunofluorescence analysis after overexpression of N-terminally myc-tagged variants of mouse HTATIP2 (**A**), PAFAH2 (**B**), SAR1B (**C**) and PDCD6 (**D**) in HepG2 cells. Signals of the myc antibodies are shown in green, while signals for the peroxisomal marker PEX14 are shown in magenta. Cut-outs of the areas highlighted by squares are magnified by a factor 1:4. While HTATIP2 and PAFAH2 nearly completely co-localize with peroxisomes, SAR1B and PDCD6 exhibit only a partial overlap with the PEX14 signals. For myc-SAR1B, Z-plane transects through the magnified area are shown to validate the fact that myc and PEX14 signals origin from the same planes of the confocal image stack (see arrowheads).

**Figure 7 cells-13-00176-f007:**
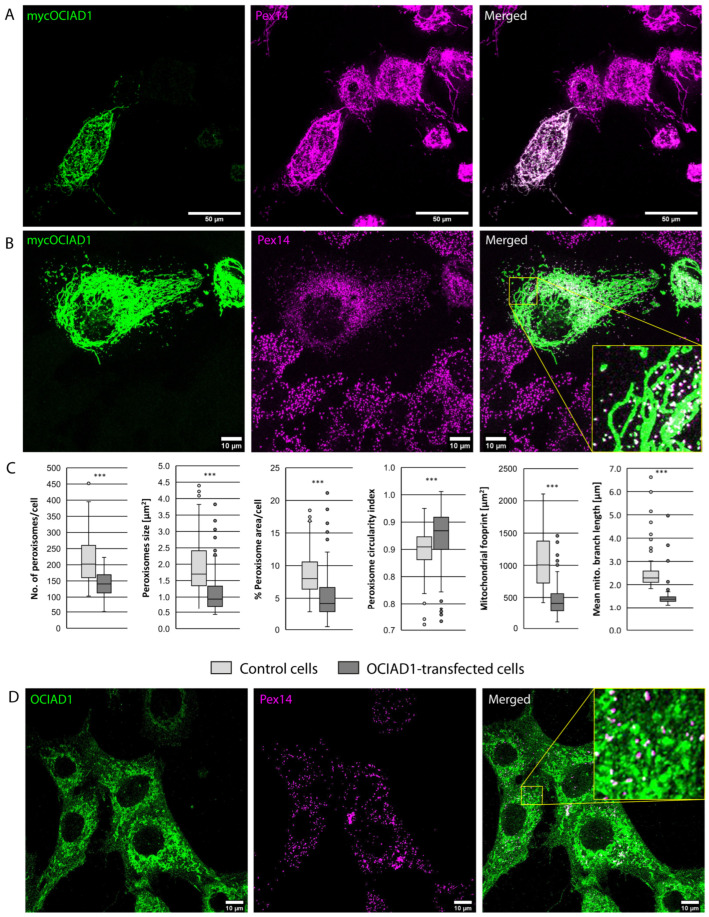
Intracellular localization of mouse OCIAD1. (**A**) Signals of overexpressed mouse myc-OCIAD1 (green) largely overlap with the antibody signals of the mitochondria marker TOMM20 (red), (**B**) the more spherical signals of myc-OCIAD1 exhibit a high degree of colocalization with peroxisomal PEX14, confirming its targeting of peroxisomes. (**C**) Effect of OCIAD1 expression on peroxisomal and mitochondrial morphology and abundance. For quantitative data, 200 control and OCIAD1-transfected cells were analyzed, using the “Analyze Particle” and “MiNA” tools from ImageJ (*** *p* < 0.01). (**D**) Immunofluorescence localization of endogenous OCIAD1 (green) in mouse embryonic fibroblasts, confirming its localization to peroxisomes (magenta). Squares in the merged images from (**A**,**B**,**D**) highlight regions magnified by a factor 1:4.

## Data Availability

All original data from the proteomics analysis are publicly available at the ProteomeXchange database under the project name “Analysis of the mouse hepatic peroxisome proteome—identification of novel protein constituents using a quantitative SWATH-MS approach” and the project accession PXD047931. The remaining data presented in this study are available upon request from the corresponding author.

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
