# Peer review of "Analysis of the Mouse Hepatic Peroxisome Proteome—Identification of Novel Protein Constituents Using a Semi-Quantitative SWATH-MS Approach"

_cells, 2024, doi:10.3390/cells13020176_

Round 1
Reviewer 1 Report
Comments and Suggestions for Authors
In this manuscript, Singin et al. present a novel and thorough proteomics analysis of mouse liver peroxisomes. Peroxisomes were purified by gradient centrifugation starting from two different “pre-fractions”: 1) the classical “light mitochondria” (LM) fraction of De Duve et al., and 2) the fluffy layer that sediments at the top of brownish pellet of the LM fraction. Purified peroxisomes were then subjected to MS analysis using a protocol that had not been used before with these organelles. Collectively, this analysis led to the identification of many potentially new peroxisomal proteins, five of which were selected for subsequent validation experiments. Two proteins ( HTATIP2 and PAFAH2) turned out to be new peroxisomal proteins. Two other (SAR1B and PDCD6) belong to the ER but probably are involved in ER-peroxisome contact sites. The fifth protein studied (OCIAD1) is a component of mitochondria and peroxisomes. This work is of high quality and the results are important for the field. I have some questions/suggestions. All are minor.
Line 49 – “During the last two…” Could be a new paragraph.
Lines 65-70 – this long sentence is confusing. Suggestion: “Hence, in this study we aimed at isolating peroxisomes from these two fractions in order to: (1) discriminate true peroxisomal constituents from contaminating proteins; (2) unravel if FLM-peroxisomes (FLM-PO) differ from peroxisomes isolated from the LM (LM-PO) in regard to their proteome; and (3) determine if proteins from attached membrane contacts zones might be specifically co-purified with peroxisomes from the two fractions.
Line 76 – the acronym “SWATH-MS”. I understand that an explanation is provided a few lines below but still it is strange to have this acronym here without a description of its meaning.
Line 142 – “carefully shaking the centrifuge tube”. Don´t the authors mean “carefully swirling the centrifuge tube”?
Line 154 – “Directly before” should be “Immediately before”.
Line 157 – When describing the vertical rotor centrifugation conditions, it would be helpful to define the rotor, speed and time of the centrifugation, and only then provide the integrated force.
Line 159 – “(Fig. 1A)” should be “(Fig. 1)”
Lines 163-167 – the sentence “LM1 and LM2 fraction were combined to the LM-PO or FLM-PO fractions” should be “LM1 and LM2 fractions or FLM1 and FLM2 fractions were combined to yield LM-PO or FLM-PO fractions, respectively”. Same for LM-TOP and FLM-TOP.
Lines 436-440. This sentence is unclear. Please rephrase.
Lines 495-496- Maybe I am missing information but PAFAH2 is a hydrolase, not an oxidoreductase. Please confirm.
Fig. 5 – The log scale is not mentioned in this figure/legend. I believe it is log10 but please confirm.
Line 736 – There is an incomplete sentence at the end of the acknowledgements section.
In supplementary tables "Log2 LM vs. FLM ratios" and "Manual cluster affiliation" proteins are color-coded but the key to the colors is very difficult to find in the first table and is not present in second table. Could the authors 1) relocate the color code in the first table from lines 320 to lines 3-5, and 2) repeat the color code in the second table?
Suppl. Fig. 1 – Please specify the volume of each fraction obtained. Otherwise, the table does not report total protein amounts, just concentration of samples.
Comments on the Quality of English Language-
Author Response
We thank reviewer 1 for his/her thorough examination of the manuscript and the helpful comments, which will certainly increase the quality of our manuscript. In detail, we included the following changes and corrections according to the reviewer’s requests:
Line 49 – “During the last two…” Could be a new paragraph.
A: A new paragraph was inserted at the proposed position.
Lines 65-70 – this long sentence is confusing. Suggestion: “Hence, in this study we aimed at isolating peroxisomes from these two fractions in order to: (1) discriminate true peroxisomal constituents from contaminating proteins; (2) unravel if FLM-peroxisomes (FLM-PO) differ from peroxisomes isolated from the LM (LM-PO) in regard to their proteome; and (3) determine if proteins from attached membrane contacts zones might be specifically co-purified with peroxisomes from the two fractions.
A: The text was modified according to the reviewer suggestion (page 2, line 65 – 70)
Line 76 – the acronym “SWATH-MS”. I understand that an explanation is provided a few lines below but still it is strange to have this acronym here without a description of its meaning.
A: Correct, we now added the explanation of the acronym in brackets (page 2, line 76)
Line 142 – “carefully shaking the centrifuge tube”. Don´t the authors mean “carefully swirling the centrifuge tube”?
Indeed “swirling” is the better term to describe the procedure. We thank the reviewer for his suggestion and changed the text accordingly (page 4, line 143).
Line 154 – “Directly before” should be “Immediately before”.
A: The text was changed according to the referee’s suggestions (page 4, line 155)
Line 157 – When describing the vertical rotor centrifugation conditions, it would be helpful to define the rotor, speed and time of the centrifugation, and only then provide the integrated force.
A: We added information on the rotor type used and the time applied for the centrifugation. However, since the time for the centrifugation was set by programming the centrifuge with the integrated force parameters given in the text, we placed the time in brackets at the end of the sentence since it might vary slightly depending on acceleration speed of the centrifuge used. (page 4, line 157 – 160).
Line 159 – “(Fig. 1A)” should be “(Fig. 1)”
A: The reviewer is right. We changed the text accordingly (page 4, line 161).
Lines 163-167 – the sentence “LM1 and LM2 fraction were combined to the LM-PO or FLM-PO fractions” should be “LM1 and LM2 fractions or FLM1 and FLM2 fractions were combined to yield LM-PO or FLM-PO fractions, respectively”. Same for LM-TOP and FLM-TOP.
- We changed the text according to the reviewer’s suggestion: “For the proteomics analysis, the highly similar LM1 and LM2 fractions or FLM1 and FLM2 fractions were combined to yield LM-PO or FLM-PO fractions, respectively. Like-wise, LM4 and LM5 – consisting of about 95% of the total protein (Fig. S1) and represent-ing a mixture of the remaining organelles applied to the gradient – were combined to yield the LM-TOP and FLM-TOP fractions, respectively” (page 4, line 165 – 170).
Lines 436-440. This sentence is unclear. Please rephrase.
We thank the reviewer for this remark. We now divided this sentence into two and added some extra information, to make its meaning, hopefully, more clear: “Only 1% and 2% of the peroxisomal proteins described to date were not significantly enriched in LM-PO and FLM-PO fractions, respectively. A closer look at the identity of these proteins (UGT1A1, ACTG1, SOD2, SLC27A2, MGST1, CYB5A, FABP1, CYB5R3 for LM; UGT1A1, RHOA, CYB5A, ACTG1, SLC27A2, MGST1, HSD3B3, CYB5R3, FABP1, RAB10, SOD2, RAB14, ALDH3A2 for FLM) reveals that these are uniformly proteins, which still lack profound experimental confirmation of their peroxisomal localization [3]. Moreover, all are with respect to UniProt database localized at a second subcellular compartment and, hence, might be found in similar concentrations in PO and TOP fractions.” (page 11, line 445 - 453)
Lines 495-496- Maybe I am missing information but PAFAH2 is a hydrolase, not an oxidoreductase. Please confirm.
A: The reviewer is correct. Indeed, PAFAH2 is a hydrolase from the serine esterase family. We changed the text accordingly (page 15, line 518 - 520)
Fig. 5 – The log scale is not mentioned in this figure/legend. I believe it is log10 but please confirm.
The scale is log2 – we added the missing information to the figure legend and apologize for this sloppiness in our data presentation (page 14, line 506).
Line 736 – There is an incomplete sentence at the end of the acknowledgements section (page 21, line 760).
A: We deleted the sentence fragment, which resulted from shifting the funding acknowledgments to the specific funding section.
In supplementary tables "Log2 LM vs. FLM ratios" and "Manual cluster affiliation" proteins are color-coded but the key to the colors is very difficult to find in the first table and is not present in second table. Could the authors 1) relocate the color code in the first table from lines 320 to lines 3-5, and 2) repeat the color code in the second table?
A: We have moved/inserted the color code to the top of each table
Suppl. Fig. 1 – Please specify the volume of each fraction obtained. Otherwise, the table does not report total protein amounts, just concentration of samples.
A: Indeed, the graph depicts the total protein amount from each fraction, but the y-axis was labeled incorrectly. We apologize for this mistake and corrected the y-axis label in Fig. S1.
Reviewer 2 Report
Comments and Suggestions for Authors
The manuscript describes efforts to characterize peroxisome protein composition. It highlights the challenges of isolating organelles in a sufficiently pure form to unambiguously differentiate between true associations vs contamination with other cellular components. A good introduction with clear rationalization for the study is provided. Generally, the methods are well detailed, and the data is supportive of the conclusions provided.
The authours have provided compelling evidence to support their conclusions. The data was discussed very objectively and caveats were presented.
I have a few specific comments.
1) The immunoblot methodology requires more detailing of the procedure regarding the sequential probing of a single blot in the absence of a strip step in between blots.
2) Several of the antibodies used seem to have rather broad specificity resulting in multiple bands on the blots. This is an unfortunate property of many antibodies, particularly if they are raised against peptide sequences. The authors should address this in their presentation of results as it impacts on the confidence of their interpretations.
3) The data should include a summary of how many samples each of the identified proteins were observed in. This could be in the form of a Venn diagram.
4) Figure 7a the mycOCIAD1 panels could be more informative if they were of lower intensity.
5)Fig. 6 legend should read PDCD6 (D) not (C)
Author Response
We thank reviewer 2 for his/her thorough examination of the manuscript and the helpful comments, which will certainly increase the quality of our manuscript. In detail, we included the following changes and corrections according to the reviewer’s requests:
1) The immunoblot methodology requires more detailing of the procedure regarding the sequential probing of a single blot in the absence of a strip step in between blots.
A: We added the following supplementary text passage to chapter 2.2 of the methods section: “In some occasions, the blot membranes were probed sequentially by two primary antibodies (see immunoblot originals in the suppl. material). To this end, after the signal of the first antibody incubation was recorded, the blot was washed overnight in PBST and the next day incubated with a second primary antibody using the same protocol. The molecular weight of the two proteins of interest was carefully selected in a manner, to avoid potential overlap of the two signals.” (page 5, line 191 – 197).
2) Several of the antibodies used seem to have rather broad specificity resulting in multiple bands on the blots. This is an unfortunate property of many antibodies, particularly if they are raised against peptide sequences. The authors should address this in their presentation of results as it impacts on the confidence of their interpretations.
A: For sure, the reviewer is right, that the result from an immunoblot analysis is always limited by antibody specificity. Since we were aware of this obstacle, we extensively described the distribution of proteins from the different subcellular locations in the fractions analyzed by MS. To make this more clear, we added the following passage to the results section: “Nevertheless, immunoblotting results are always limited by the quality of the primary antibodies available and, hence, a more extensive characterization of the individual fractions on a large scale by MS proteome analysis can be used to confirm the results gained by immunoblotting” (paragraph 3.1, page 9, line 388 – 391). Additionally, we added marks for the MW standards to the file depicting the original immunoblot images in order to indicate the molecular weight of the bands presented in Fig. 2.
3) The data should include a summary of how many samples each of the identified proteins were observed in. This could be in the form of a Venn diagram.
A: We thank the reviewer for this meaningful suggestion. We added Venn diagrams illustrating the conformity of the individual sample replicates now shown in a novel Fig. S1B. Moreover, we added a list of the individual protein quantifications in the replicates of the different samples to the Suppl. data file (Excel sheet “Protein quant. – replicates). Additionally we added a small text passage into the main text according to the novel information presented: “Generally, protein quantifications are based on at least 3 replicate values ≥ 90 % of the analyzed proteins (Fig.S1B), underlining the high data completeness in SWATH-MS compared to classic shotgun proteomics approaches. To further evaluate the data, function and subcellular localization of the individual proteins were annotated according to the information available at UniProt” (page11, line 421-426).
4) Figure 7a the mycOCIAD1 panels could be more informative if they were of lower intensity.
A: The mycOCIAD1 panel in Fig. 7A was replaced by an image with lower intensity. Additionally we switched the color from the PEX14 panel to magenta to fit the image to a color spectrum suitable for color-blind individuals.
5) Fig. 6 legend should read PDCD6 (D) not (C)
A: The legend was corrected accordingly (page 16, line 564).